# Becoming Cat People: Animal-like Human Experiences with a Sensory Augmenting Whisker Wearable

**Leave Authors Anonymous**
for Submission
City, Country
e-mail address

**Leave Authors Anonymous**
for Submission
City, Country
e-mail address

**Leave Authors Anonymous**
for Submission
City, Country
e-mail address

**Leave Authors Anonymous**
for Submission
City, Country
e-mail address

**Leave Authors Anonymous**
for Submission
City, Country
e-mail address

**Leave Authors Anonymous**
for Submission
City, Country
e-mail address

**Leave Authors Anonymous**
for Submission
City, Country
e-mail address

## ABSTRACT

Humans have a natural curiosity to imagine what it feels like to exist as someone or something else. This curiosity becomes even stronger for the pets we care for. Humans cannot truly know what it is like to be our pets, but we can deepen our understanding of what it is like to perceive and explore the world like them. We investigate how wearables can offer people animal perspective-taking opportunities to experience the world through animal senses that differ from those biologically natural to us. To assess the potential of wearables in animal perspective-taking, we developed a sensory-augmenting wearable that gives wearers cat-like whiskers. We then created a maze exploration experience where blindfolded participants utilized the whiskers to navigate the maze. We draw on animal behavioral research to evaluate how the whisker activity supported authentically cat-like experiences, and discuss the implications of this work for future learning experiences.

## Author Keywords

Wearables; biosensing; empathy; exploration; exploitation

## CCS Concepts

•**Human-centered computing** → **Human computer interaction (HCI); Mixed / augmented reality; Interactive systems and tools;** *HCI design and evaluation methods;* •**Social and professional topics** → **Computing education;** *Adult education;* •**Applied computing** → **Interactive learning environments;** •**Hardware** → *Biology-related information processing;*

© 2020 Copyright held by the owner/author(s). Publication rights licensed to ACM.
ISBN 978-1-4503-6708-0/20/04. . . $15.00
DOI: https://doi.org/10.1145/3313831.XXXXXXX

## INTRODUCTION

Posthumanist philosophies characterize the human body as "the original prosthesis we all learn to manipulate" [22], and suggest the idea that augmenting or substituting aspects of this prosthesis is the normal progression for humanity. Technology allows humans to enhance senses that may be impaired, and to extend our bodies with added senses beyond what we would otherwise be biologically limited by—giving humans the ability to improve their quality of life [17, 18]. "In short, we are cyborgs" [21]. Scholars have investigated how immersive virtual environments can enhance social perspective-taking [20, 47], and computer-augmented, embodied perspective-taking has been shown to encourage a productive "learning stance" [33] and to enhance both conceptual learning and engagement [15, 34]. Some environmental education scholars [40, 54] and indigenous educational scholars [5] have suggested that building relational ties to non-human actors in nature may contribute to environmental and biology education. In a few cases, educators have asked learners to take on the embodied experiences of insects such as bees [15] and animals such as polar bears [37]. Danish found that children enacting a computer-augmented pollination activity embodying the roles of bees helped them learn nuances of individual and aggregate bee behavior; Lyons and colleagues found that wearable polar bear paws that simulated the feeling of traversing melting polar ice enabled people to show an empathetic understanding of the impacts of climate change.

For many people, the most common experience they will have with entities who have different sensory capabilities is through

everyday interaction with pets or neighborhood animals. For example, in noticing that our house cat is navigating a dark space where we would likely bump into something, we may recognize the limits of our own senses and consider how our pets' experiences are both similar to and different from our own. Our work explores how embodied technology can mediate human experiences in ways that offer people opportunities to explore and relate to the animal-like behaviors of their pets.

We present the design of a cat-inspired whiskers wearable, the Whisker Beard, and an embodied navigation activity that provided a firsthand perspective-taking experience for participants curious about what it might be like to have whiskers. In addition, we discuss our philosophy of what it means for an animal-imitating experience to be authentic and we present the evaluation framework we used to understand how our whiskers activity encouraged participants to behave like cats.

Our study addresses two research questions:
**RQ1**: In what ways can we create technologies and environments that remediate human experiences to be like those of non-humans?
**RQ2**: What are humans' impressions of these technologically remediated experiences?

In this paper we describe (1) the design of a sensory augmentation whiskers wearable; (2) the creation of a maze exploration activity for testing the experience of wearing whiskers; (3) our analysis methods for evaluating the authenticity of an animal-like experience; and (4) outline opportunities to extend this work, as well as discuss the implications of it.

### Motivation
Our technology, experience design, and analysis are motivated by a desire to re-shape science and science education in light of feminist critiques and visions of those fields. Whereas science and science education today tend to emphasize distance, objectivity, and dispassion, we strive to create spaces for discovery and learning that also include closeness, subjectivity, and emotion [26] and that thereby enable learners to author identities in science [7, 25] based on more engaged and connected ways of knowing [8]. The overall approach of the project is to investigate how affection for and curiosity about pets can catalyze scientific investigations and engineering for young people and their families that are based on empathy and perspective-taking, both in and out of schools. We see value in using wearable and mixed-reality technologies to provide humans with the ability to experience the lives of other beings. In particular, this would allow humans to experience what their pets' senses might be like, and thereby facilitate learning experiences that encourage empathy and perspective-taking. Once we can support interspecies perspective taking, we then wish to encourage participants to conduct scientific inquiry within that intersubjective sensational realm – a realm which the German biologist Jakob von Uexküll called "Umwelt" [14].

This agenda has the potential to unify efforts to advance scientific and social-emotional education. Humans have strong emotional attachments to their pets and these human-animal bonds coincide with higher amounts of empathy [4]. This

can motivate people's curiosities about their pets' lives and experiences [4, 52, 60].

One technological and philosophical challenge involved in pursuing our agenda is the intrinsic disconnect between humans' experiences and those of other species. People only know what it is like to be human, and are unable to know on a phenomenological level what it is truly like to be their pets [58]. Though people cohabitate with their pets, on a biological level they see, hear, smell, taste, and experience the world differently from them— ranging from slight variations of senses to things incomprehensible and alien like sonar and magnetic field detection. Despite this challenge, most pet owners believe that their pets feel something, even if they cannot fully understand what they feel [4]. People naturally evaluate animal behavior and experience through a human lens. In a thought exercise of imagining what it is like to be a bat, Nagel highlights the mind-body problem he encounters: "...I want to know what it is like for a bat to be a bat. Yet if I try to imagine this, I am restricted to the resources of my own mind" [42].

While we will likely never solve the mind-body problem of humans truly understanding their pets' experiences, we can address the "body" aspect of the problem and use it to drive thought experiments about the lives of pets. These thought exercises can give people a better sense of animals' lives by acknowledging biological differences that many people have about how their pets see, hear, and feel the world [32, 42].

In this study, we present the results of an early step toward feminist science and science education: the design and evaluation of a wearable device meant to offer humans the experience of exploring an environment using whiskers. This first step is intended to both elucidate how new technologies can offer transspecies sensory experiences, and to show how we might assess the validity of those experiences (i.e., the extent to which they immerse humans in the reality of another species) by comparing humans' behaviors in the new technologically-mediated umwelt with those whom the umwelt natively describes.

## RELATED WORK

### Wearables, Mixed Reality, and Sensory Augmentation
Wearables and mixed reality technologies have been applied in a wide variety of domains ranging from educational contexts [15, 19, 33, 43], medical settings [28], natural environments [35], performances [3, 65], and museums [37, 64]. The resulting hybrids of wearables and mixed reality technologies are sensory augmentation devices. Sensory augmentation devices give humans the ability to experience phenomena that they are physically unable to process, as well as some phenomena that are simply unnatural to the human experience [51, 56, 61]. The field of sensory substitution and augmentation enables humans to use existing senses to substitute for the ones that they cannot experience, and to augment senses they already have to make them more powerful. For example, sensory substitution has been applied as a method for offering hearing and/or visually impaired people additional senses to substitute for the one in which they have an impairment. These technologies range from devices that can be implanted within the body, sending direct signals to internal mechanisms in the brain, to

external substitute devices that provide tactile feedback on the surface of the skin [2]. In one example of tactile feedback, researchers developed a non-invasive "vibratory vest" for deaf and hearing impaired individuals that processes auditory information and converts it into vibrotactile feedback on the wearer's torso [44, 45]. Our work builds on approaches that use these technologies to provide physical, sensory feedback experiences for the wearer.

## Wearables and Education

Wearables have the ability to provide hands-on, interactive, and embodied learning experiences. There is research to suggest that wearable technologies in the classroom can increase engagement and improve student attitudes towards STEM activities [6, 30]. One of the pedagogical affordances of wearable technologies is the ability to gather contextual information from a firsthand account [10]. Some uses of wearables in education focus on understanding and quantifying the self, such as: calculating our heart rates, monitoring how many steps we have taken, and providing feedback about our emotional states [16, 23, 53]. Other approaches embed the quantifying potential of wearables in scientific inquiry and discourse [29].

In contrast, our research focuses on how wearable technologies can help people understand experiences beyond the self, more specifically, the potential for wearable technologies to foster empathy by helping humans understand the experiences of others. Incorporating empathy and perspective-taking into scientific question-asking is deeply rooted in feminist educational theory and practice [7, 25]. These practices offer a more inclusive view of what it means to actively participate in science, and promote empathy as a valued part of scientific discovery [63]. An example of a wearable device aimed to motivate empathy is a series of mushroom foraging tools designed to build more intimate relationships between humans and the environments they are probing by connecting them physically to the environment [35]. Nobel laureate Barbara McClintock cited her ability to get "a feel for the organism" as influential in supporting her discoveries in the area of maize cytogenetics [26]. McClintock has said, "I know my corn plants intimately, and I find it a great pleasure to know them," a sentiment we believe could inform future wearable-mediated science education.

## Animal Navigation Behaviors

When interacting with non-human animals, it is easy to notice how their sensory capabilities differ from humans'. For example, people may notice their cat hear or see something without hearing or seeing it themselves. In addition to animal companions' sharper senses, people may also notice qualitative differences in how they experience the environment, such as when their pets deftly navigating through small spaces. People may also notice and wonder about the presence of physical characteristics that humans lack, such as whiskers or purring behaviors. Our work in this paper is motivated by human curiosity about these differences between species and explores how taking on the physical or perceptual characteristics of an animal helps people to understand it. In this study, we specifically focus on how participants' behavior compares to

navigation and foraging techniques that animals exhibit in the wild and in controlled laboratory maze experiments.

An animal's ability to gather environmental information and make decisions in real-time is key for its survival. Animals must constantly make decisions and consider tradeoffs during activities like foraging for food, avoiding predators, and searching for a mate. These trade-offs describe what is called the exploration-exploitation problem, a key idea in organizational learning and animal behavioral science. A common definition describes exploration as the process of randomly searching an unrefined and unexplored area, and exploitation as the process of searching a more refined area for some reward or resource [41]. The less information an animal has about an environment, the more likely it is to be exploring rather than exploiting [41]. Researchers argue that optimal search strategies for animals include alternating patterns of quick movement and searching [9].

In the process of exploring and exploiting an environment, animals exhibit "egocentric" and "geocentric" navigational strategies. In an egocentric strategy, an organism uses previously acquired information, either from memory or another internal mechanism, to move along a path. In a geocentric strategy, an organism uses cues and real-time feedback from the environment to continuously reorient itself along a path [49]. Animals rely heavily on geocentric strategies to explore and search unfamiliar areas; it is too difficult to primarily rely on internal mechanisms when moving through new territory [13]. The term thigmotaxis describes the behavior of either moving toward or away from touch stimulus; an example of this behavior is "wall-hugging" in animals, which is the tendency to avoid open areas and stick to the perimeter of an environment during navigation [59]. Wall-hugging behavior in animals is commonly tested in laboratory maze environments, where high levels of wall-hugging can suggest anxiety or fear in an animal. Therefore, wall-hugging is commonly seen as a strategy that animals invoke when exploring a new and unfamiliar environment [57, 62]. For example, mice spend, on average, 74% of their time hugging a wall in their first five minutes exploring an unfamiliar environment, and about 65% of their time doing so by thirty minutes [57].

## WHISKER DESIGN AND IMPLEMENTATION

We created a sensory augmentation device, the Whisker Beard, that provides a physical simulacrum of what it is like to have whiskers. We wanted our whiskers to be functionally and aesthetically reminiscent of cat whiskers, including the appearance and placement of the whiskers and the physical properties of the materials. In addition, we wanted the functionality to be authentic in the sense that a person wearing it would use the whiskers to enact behaviors similar to animals'.

## Whiskers, Animals, and Technology

Whiskers, formally known as vibrissae, are rigid but flexible hairs that provide tactile sensory feedback in many mammals and rodents. Ahl describes a number of characteristics and roles for whiskers; whiskers differ from hair in that their follicles are much thicker than normal hair follicles, and that whiskers are sensitive and send signals to the brain. Whiskers

are commonly localized to facial regions of animals, but can also be found on other parts of the bodies of animals. They are essential to survival, monitoring, communication, and aggression. They help serve as a sensory substitute for animals with poor close-sightedness; cats have difficulty seeing objects very close to their faces, so whiskers help them sense objects close to them, as well as protecting their face from harmful objects [1].

A project close to our vision is a helmet that uses infrared sensors to detect close-by objects and provides vibrotactile feedback to the wearer; like our project, [27]'s design was inspired by mammals' abilities to use whiskers to detect their surroundings. Our project and theirs are similar in that they are whisker-inspired designs for sensory augmenting wearable devices, however our goals differ in that we are not trying to offer an efficient solution for humans to navigate in low-light conditions, but rather to provide perspective-taking experiences for wearers to experience what it is like to have and use whiskers.

## Whisker Design

The version of the Whisker Beard used in this study has a total of four whiskers, two for each side of the face. In order to mimic the genal (cheek) whiskers of the cat, our whiskers attach to the sides of two acrylic cheek plates that rest 2-3 cm above the skin of the wearer, therefore the whiskers protrude from the sides of the wearer's cheeks close to the mouth (see Figure 1). Each whisker's total length (38.5 cm) is approximately the average human's shoulder width (39 cm) [38]. We chose this length in order to make the whiskers extend beyond the shoulders of an average sized person, thereby imitating the appearance and functional reach of cats' whiskers. Figure 1 shows a researcher wearing the Whisker Beard.

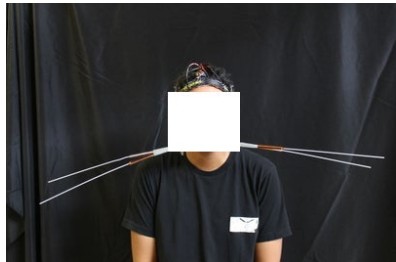

**Figure 1. A researcher from our team wearing the Whisker Beard (photo anonymized for submission).**

The whiskers are composed of three flexible components: a flex sensor that detects the deflection that the whiskers are receiving, polystyrene strips that extend the usable length of the flex sensor significantly and returns the flex sensor to a neutral unbent position, and Sugru, which is a moldable silicone glue that holds the polystyrene strip and flex sensor together. In addition, a small connector at the end of the whisker attaches to the cheek (see Figure 2).

Our device detects when the wearer brushes their whiskers against a surface. When the whiskers are bent the change in resistance is measured and converted to a voltage. The

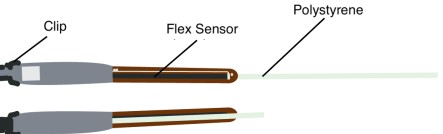

**Figure 2. Material makeup of our whiskers.**

calculated voltage measurement is then used to set the pulse-width-modulation signal to the vibration motor control. The device provides vibrotactile feedback to the wearer by varying the intensity of vibration proportional to the bend angle through an array of vibration motors on the scalp, inspired by the work from [45]. Each motor is coupled to one whisker. We chose to place the vibration motors on the scalp so the vibrotactile sensation would be felt on the wearer's head, and so the motors could be placed far enough apart to allow for better point discrimination (making it easier for the wearer to identify which specific motors are vibrating) [24]. The whiskers sense bidirectionally in order to better mimic a cat's actual sensory perceptions when approaching and backing out of different confined locations. The current hardware design allows for bidirectional sensing, but the software does not explicitly indicate direction of the whiskers' bends through haptic feedback, however, the participant can feel the direction of tension on the wearable.

## MAZE ACTIVITY AND WORKSHOP DESIGN

In order to study how the Whisker Beard can support animal-like environmental exploration, we built a human-sized cardboard maze (4m x 6m) for our participants to explore while wearing the whiskers wearable. Mazes are a common tool in behavioral research involving rats and smaller mammals [12]. Our hypothesis was that participants would be able to use the whiskers as a workable form of sensing during maze navigation. We wanted participants to rely on the whiskers as heavily as possible and other senses as little as possible. Therefore, we chose to blindfold the participants since vision is the dominant sense for non-visually impaired humans for gathering environmental information [50].

We designed the maze to promote enough confusion for participants so they would need to use feedback from the whiskers to guide them. We did so in order to observe a more honest range of animal behavior. Before creating the human-size maze, we created a series of small-scale prototypes. We used these to consider different possible maze routes as well as what kinds of obstacles would promote explorations of the whiskers' affordances. We prototyped obstacles that would encourage participants to enact cat-like behaviors such as rubbing one's face on a surface, and having to back out of narrow spaces. We came to the conclusion that experimenting with unfamiliar topography was the best way to create an experience that removes participants from a human sensory experience within an exploration task. Therefore, the final maze design included multiple dead ends, corridors of different widths, vertically hinged flaps, and cutouts in the walls. In addition, due to the bidirectionality of the whiskers, we designed our obstacles such that wearers would mostly need to rely on the horizontal

changes in the whiskers' shapes. Figure 3 shows a photo of the finished cardboard maze, and Figure 4 shows a digital rendering of an aerial view of the maze.

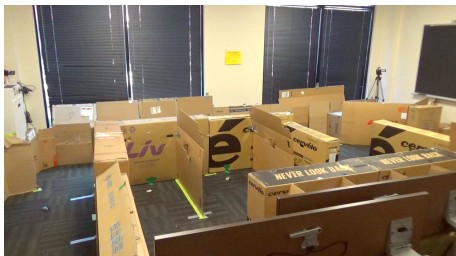

**Figure 3. The final human-scale cardboard maze.**

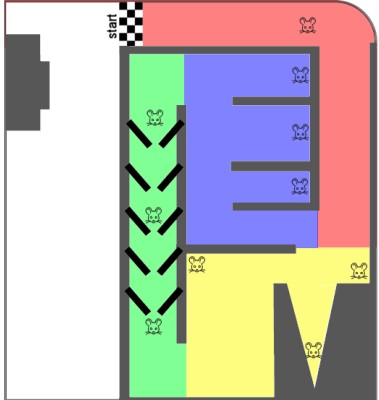

**Figure 4. An aerial map of the maze with four colors (red, blue, yellow, and green) to indicate particular sections of the maze. The mouse icons indicate where we placed the toy mice.**

We placed 10 toy mice throughout the maze to induce exploration and exploitation behavior. We gave participants this task in order to give them a survival-oriented goal to collect as many resources as possible from an unfamiliar environment in a limited time frame to strengthen the worldview of being an animal.

## METHODS
We investigated how participants used the Whisker Beard to navigate the maze, and in our analysis we focus on what ways participants did or did not exhibit animal-like behaviors during the experimental activity, and how they reacted to the experience of cat-like sensory augmentation. To do so, we evaluate how participants behaved like cats, as well as what participants' impressions were about the activity. We draw on work from animal and biological sciences, particularly key areas of research related to navigation and animal behavior. In addition, we highlight moments from participant discussion that illustrate ideas and questions they had after the experience.

### Recruitment
We recruited six undergraduate college students who lived on campus in the dormitory that we constructed the maze in. Due to the circumstances of living in a dorm room, none of the participants currently owned a pet, however three of the participants had previously owned a cat, or still had a cat at

home, and the participants who had not owned a cat had owned a dog and also knew family and friends who have had cats.

### Experimental Procedure
All six participants completed the maze activity on the same day in subsequent 20 minute sessions. As participants arrived for their sessions, we greeted them individually in a separate room. There, we assisted each participant in attaching the Whisker Beard to their face. We spent a few minutes explaining how the device works and gave participants time to learn the correlation between whisker bend and vibration location and intensity. We did this by individually flexing each whisker while the participant was wearing it (prior to blindfolding them) and asking them to describe if they could feel the changes in the vibrotactile feedback. Once participants were familiar with the wearable, we explained the maze procedure to participants, and brought them – blindfolded – into the maze room. We replaced all 10 toy mice in the same locations before the start of each participant's session (see Figure 4).

We gave each participant a maximum of 10 minutes to crawl through the maze and explore, and to collect as many toy mice as they could. We asked participants to think-aloud [31] during the maze so we could have a better sense of their thought process throughout the activity to both validate our coding of the data, as well as collect information about participants' impressions of the activity.

After each participant completed the maze activity, we asked them to refrain from discussing the experience with other participants until everyone was finished. We had them each fill out a worksheet to provide responses to questions like, how did it feel to navigate using whiskers, and what moments stood out to you when using your whiskers to navigate the maze? We selected open-ended questions such as these to gather information about participants' experiences on an individual level and to get their initial impressions right after the activity.

After all participants were finished with the activity, we reconvened as a full group for a facilitated discussion. We began the discussion with three prompts on the board: (1) "How did you feel when you first put the whiskers on?" (2) "What particular moments in the maze stood out to you?" and (3) "Other thoughts?" We asked participants to write their thoughts about these prompts on sticky-notes and place them on the board. Once participants completed this, we used the responses to facilitate discussion among the whole group.

### Analysis
In order to investigate how the wearable and maze activity mediated participants' experiences in an animal-like way (RQ1), we analyzed participants' interactions, behaviors, and commentary throughout the activity through coding of the data and analysis of their movement. We assessed interrater reliability by having each coder annotate a map of the maze using the process shown in Figure 7. We compared each annotation on both maps. If the coders identified an action as a different event type as shown in Figure 6, or if one coder identified an event at a particular time and the other did not identify an event, these were marked as disagreements. We then added up all the agreements and disagreements to calculate with

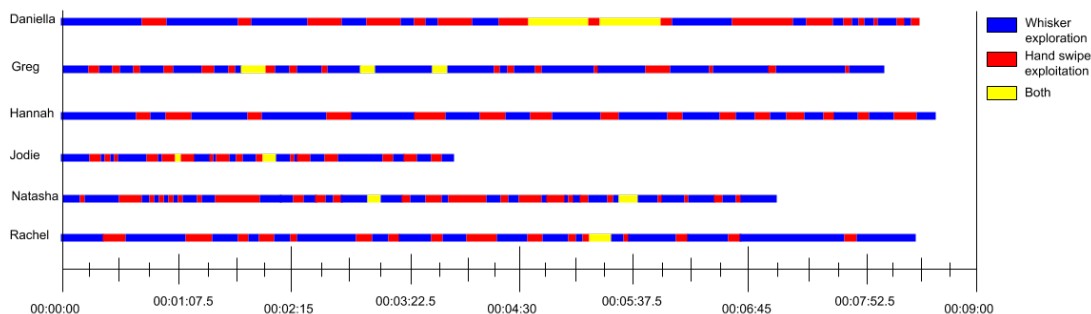

**Figure 5. A visualization of participants' usage of their whiskers and hands over time. Note that Jodie's time is much shorter than other participants due to the fact she moved through the maze very quickly and returned to the starting point after about four minutes.**

Cohen's Kappa. For example, to assess agreement on where "whisker interactions" occurred, we looked at where the raters identified whisker interactions on their maps, and counted all matching identifications as agreements, and all discrepancies as disagreements. We did this for all symbols in the key and totaled the results. After reaching an inter-rater reliability of k=0.62, which indicates substantial agreement [39], the two researchers coded the rest of the data in parallel.

To analyze the paths of participants and their interactions, we captured audio and video recordings of participants' movements and interactions in the maze from five cameras: we placed four cameras in different positions along the maze's perimeter, and a researcher held one camera and followed the participant as they moved through the maze. We analyzed the videos to produce content logs [36] and time stamps of different events. In addition, we used the videos to manually generate aerial maps that depict participants' movements and interactions through the maze. Figures 6 and 7 illustrate the orientation, position, and scale of the participant, as well as depict participants' interactions with their whiskers and hands at a given time.

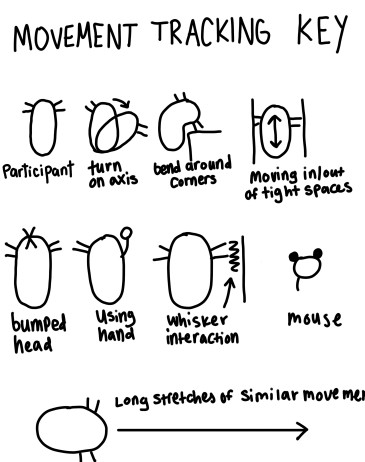

**Figure 6. Our key for generating the aerial maps of participants' paths.**

In addition, we denoted four subsections of the maze that we thought represented the different types of obstacles and areas (see Figure 4). The red zone is where the maze starts and contains two straight hallways. The yellow zone contains

a large area of open space, as well as a wide opening that narrows to a dead end. The blue zone contains a straight hallway that opens into three smaller corridors. The green zone is a long hallway with cardboard flaps hanging on either side that participants had to push through.

In Shapiro and Hall's museum mapping work, they created aerial maps to illustrate museum visitor movement and engagement to understand how the space created by the gallery facilitates learning [55]. We apply their mapping methods in our study in order to identify how participants interacted with the space with the Whisker Beard and how their interactions and movement compared to animal-like behavior (see Figure 7 for examples of the paths we generated for analysis and refer to Figure 6 for our key).

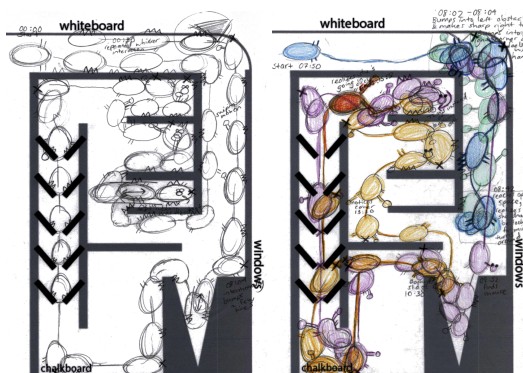

**Figure 7. Two researcher-generated scans of different participant maze maps. Refer to Figure 6 for our key. The map on the right contains color so that we could draw overlapping paths and keep track of the order in which it occurred (such as when participants doubled back to specific areas).**

In our analysis, we define moments of primarily whisker interaction as exploration, and moments of hand-swiping interactions as exploitation. We encouraged participants to solely rely on their whiskers to navigate the maze, and to only use their hands when they needed to collect mice. Because of this, we find the distinction between whisker use versus hand use as a reasonable indicator of exploration-exploitation behaviors. Finally, we coded moments of geocentric and egocentric strategies using the perspectives defined earlier, and coded wall-hugging to be moments where a participant makes con-

| Participant | Exploration (%) | Exploitation (%) | Explore interval length (sec) | Exploit interval length (sec) |
|---|---|---|---|---|
| Daniella | 68.4 | 44.8 | 17.9 | 12.4 |
| Greg | 81.3 | 24.2 | 14.8 | 5.2 |
| Hannah | 75.9 | 24.1 | 23.2 | 8.9 |
| Jodie | 57.0 | 55.0 | 9.1 | 6.2 |
| Natasha | 62.5 | 41.1 | 7.2 | 5.2 |
| Rachel | 77.0 | 25.8 | 18.7 | 6.6 |
| Average | 70.3 | 35.8 | 15.2 | 7.4 |

**Table 1. Participants' time spent exploring and exploiting. The percentages sometimes add to more than 100% because of overlap in moments where participants were exhibiting both behaviors at once.**

tinued contact with their body and or whiskers against the wall.

## RESULTS
We separate our results into four categories of participants': (1) patterns of exploration and exploitation and how they related to whisker use; (2) geocentric and egocentric strategies; (3) wall-hugging behaviors; and (4) reflections on the experience. The first three are aimed at providing information about the authenticity of the animal-like behaviors that participants exhibited (RQ1), while the fourth addresses what impressions people had about the wearable and the experience (RQ2).

### Exploration-Exploitation
Participants alternated between periods of long exploration and relatively shorter exploitation. In Figure 5 we illustrate the exploration-exploitation search behaviors of each participant (listed with pseudonyms). Blue segments denote periods of navigation where participants were primarily relying on their whiskers to move around. Red segments denote moments where participants moved their arms across the floor in order to search for and obtain mice. Yellow segments are places where we coded participants as using both exploration and exploitation, which was rare because crawling and swiping is challenging.

In Table 1 we show the breakdown of the amount of time participants spent exploring and exploiting the maze. Participants spent an average of 70.3% of time exploring and 35.8% of time exploiting. Participants switched back and forth between exploration and exploration, with an average exploration interval length of 15.2 seconds and an average exploitation interval length of 7.4 seconds. On average, participants caught seven out of the ten mice.

During periods of whisker exploration, participants enacted three common whisker techniques that primarily occurred in the long, straight parts of the maze (red zone in 4). In Figure 8 we illustrate what these three techniques look like.

In one technique (A) the wearer constantly drags the whiskers against the wall as they move. In the second (B) the wearer alternates between motion and pausing to brush the whiskers against the wall, and in the third (C) the wearer moves between walls alternating between the sides of the whiskers they use. All six participants enacted at least one form of A and B during the maze, and two participants utilized the C technique as well. The two participants that enacted all three techniques both have pet cats at home.

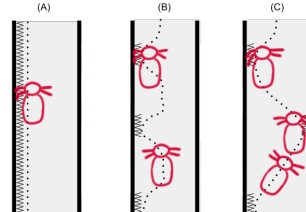

**Figure 8. Three whisker-environment interaction techniques. The dotted line indicates the participants' path, and the zig-zag line indicates whiskers rubbing on the wall.**

### Geocentric and Egocentric Search Strategies
Participants used a variety of geocentric and egocentric search strategies as they navigated the maze. Table 2 shows the result of this categorization and demonstrates the different geocentric and egocentric strategies that participants enacted, with examples and quotes to illustrate how they used particular environmental feedback.

We found that all six participants relied on enacting geocentric strategies throughout the entire activity, and used egocentric strategies more sparingly. During the think-aloud all six participants made comments about using touch and sound feedback from the whiskers, as well as tactile feedback from their hands and body. Only two participants made comments about attempting to use egocentric strategies, with one participant commenting that he was able to create an internal map during the maze (which he later admitted was incorrect after seeing the maze).

### Wall-Hugging
All six participants exhibited wall-hugging behavior (thigmotaxis) as they moved through the maze blindfolded.

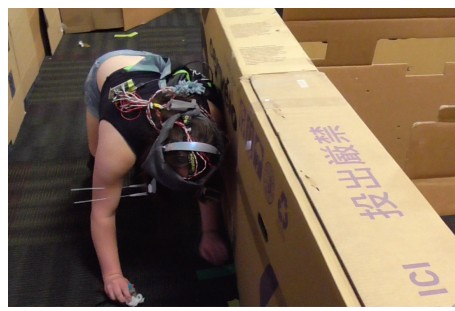

**Figure 9. A participant demonstrating wall-hugging behavior.**

| | Feedback Mechanism | Quotes/Description |
|---|---|---|
| Geocentric | Vibrotactile feedback from the whiskers | "I'm going into a narrower space, because there are whiskers [activated] on all sides." |
| | Tactile feedback from hands and the body | Participants swiped their hands searching for mice. |
| | Sound of the whiskers touching the walls | "The navigation is also noise of the whiskers touching things." |
| Egocentric | Memory of the space from an earlier point in time | "I know I'm in the center of the room, because I've been in this room before with this outlet on the floor" |
| | Memory of revisiting the same part of the maze | "I think I'm going in circles" |
| | Internal recall of physical orientation | "[I was] thinking about my previous and next moves, as well as an internal map." |

**Table 2. A breakdown of geocentric and egocentric navigational strategies that participants used.**

Five of the six participants spent over half of their time in the maze hugging the wall, with one participant hugging the wall for 97.3% of her maze time. On average, participants spent 68.6% of their time in the maze hugging the wall (min=34.6, max=97.3, standard deviation=20.87). Participants' tendencies to hug the wall varied, but interestingly, the participant who hugged the wall the least had a higher tendency than others to bump into obstacles head first (14 times). Participants frequently wall-hugged while exploring using whisker technique A from Figure 8.

**Participant Reflections**

Participants reflected on their experiences in the whisker activity through individual worksheet responses and a full group discussion. During their reflections, participants commented on the physical experience of the whisker activity, and drew connections to phenomena they have experienced with their own pet cats or others' cats.

In response to the question, "How did it feel to wear the whiskers?" participants described both the physical sensation of how it felt to have whiskers, and the functional use of the whiskers and their ability to adapt to it. Responses that described the feeling of adapting to the whiskers include:

- "As I got used to them, the whiskers started to become a part of me."
- "I liked having another sense. They got much easier to use as I played around with angles and pressure on the sensed surfaces."
- "It was an easy way to 'see' side to side. Times in empty space though stood out, with nothing to feel in front of me made me more cautious"
- "The first time I went through a small space and both whiskers activated [while] having to back out...stood out to me"
- "It was a bit odd at first but I quickly got used to them. It was kinda nice having an additional aid apart from the feeling in my hands and feet."

Responses that described the more physical feeling of having vibrating whiskers include:

- "I had to push through the flaps and it was almost over-whelming with vibrations."
- "It was an interesting feeling having the vibration kind of tickle. When I was really close to something it was also kind of shocking and made me want to back away."

Most of the above responses to the question of how it "felt" focused on the overall experience of using the whiskers to navigate, as opposed to the specific physical feeling of the vibrations on the skin.

In addition, we were curious whether participants would think about their own pet cats, or other cats that they know. We asked, "Did you think about your own pet cat while you were in the maze? If so, what did you think about?" and participants responded in ways that included perspective-taking commentary that were empathetic. Their responses included:

- "I thought about my friend's cat and how when she was five, she cut off the cat's whiskers thinking they were long hairs. For a good week the cat had to stumble around, falling over, and running into things often."
- "How cats sometimes bump their heads into things then back-up confused. I could sympathize."
- "Yes I did, I thought about how the vibration was a little like how they would use their whiskers. I also thought about how hard it would be to navigate without her whiskers"

Most of the empathetic responses show participants acknowledging that it would be difficult to navigate as a cat stripped of its whiskers, similarly to how it was difficult for them to navigate without their sight. Participants described moments of feeling disoriented when they entered the large open area of the maze (yellow zone in Figure 4), and during moments of technical difficulties when a whisker fell out. According to participants, the open areas were disorienting because they lost their sense of physicality and location, suggesting some understanding of, and potential for empathy with, animals' thigmotactic strategies.

**DISCUSSION**

The results of the Whisker Beard and maze activity show examples of participants exhibiting behaviors and strategies similar to those that animals perform when searching for re-

sources in unfamiliar environments. We separate our results into discussions about their physical behaviors and strategies, as well as their impressions of the experience.

As depicted in Figure 5 and Table 1, as participants explored the maze, they alternated between periods of explorative and exploitative behavior as they switched between using their whiskers and using their hands. Participants spent, on average, a longer amount of time exploring and moving through than maze than they spent hand swiping to look for mice. These results are in line with animal foraging behaviors [9, 41]. Benichou et al. says that animals searching for resources switch between periods of motion and periods of searching. In addition, their work shows that intervals of exploration tend to be longer than intervals of exploitation. This aligns with the amount of time our participants dedicated to these behaviors [9]. While we cannot claim that participants would not have enacted similar exploration-exploitation behaviors without whiskers, we can say that the behaviors that they enacted with whiskers were in line with foraging behaviors. Interestingly, several of the participants made use of the whiskers in ways that strikingly resembled cat behavior, as depicted in Figure 8. As participants moved down long passages, some used their whiskers to gauge the width of the passage by moving back and forth brushing each side of their whiskers on the opposing walls. This demonstrates that participants used the whiskers to enhance their spatial awareness, one of the supposed evolutionary factors behind the presence of whiskers [11]. We noticed that when participants used techniques B and C, they mimicked the behavior of cats who rub their olfactory face glands on objects to mark their scent, as well to get a sense for the physical properties of a specific object [48]. While this behavior in cats is not necessarily used for navigation purposes, it is used for gauging the size and shape of an object. Participants did this in order to look for hidden passageways and moveable obstacles.

Our observations of participants' geocentric and egocentric behaviors provided us with a fuller picture of how participants used the whiskers in tandem with other strategies during the activity. Participants relied on the vibrotactile feedback from the Whisker Beard in determining their path of movement through the maze. In addition to the vibrotactile feedback, we found that participants also relied on the sounds the whiskers made as they brushed against the maze's cardboard surfaces. We validated this observation through think-aloud commentary that participants provided throughout the maze, and through post-maze group discussion. The fact that participants relied on additional tactics beyond the vibrations is not an inauthentic outcome, but rather a reasonable one. Participants' use of different egocentric and geocentric tactics is naturally aligned with how animals navigate the world—getting the most information from their environment by whatever means are accessible to them [41]. The blindfolded maze procedure afforded participants the ability to experience the Whisker Beard in an unfamiliar environment. As expected, due to the unfamiliarity of the environment, participants relied on more geocentric strategies. These results are in line with animal navigation research which suggests that egocentric strategies are too difficult to use when exploring new terrain, and therefore animals rely more heavily on geocentric strategies to gather real-time physical feedback [41]. In time, participants who revisited areas of the maze began to recognize their surroundings, which led them to use internal recall from their memory to identify their approximate position; however, because they were blindfolded they still had to rely on geocentric strategies as well.

Unsurprisingly, participants told us that being blindfolded and losing their sense of sight was disorienting for them; sight is one of humans', and cats', dominant senses for obtaining information about an environment [50]. Participants described the open-space areas of the maze as "disorienting" and tended to try to find a wall as quickly as they could to reorient themselves. The level of consistent wall-hugging that participants exhibited is in line with experiments where increased levels of anxiety correlated to higher levels of thigmotaxis. Usually, animals' tendency to hug the wall would decrease as an experiment went on, except in circumstances where animals do not have enough time to fully process their environment [57]. In our experiment, blindfolding the participants made it challenging for them to produce an accurate internal map of the space, leading them to continuously avoid open areas and rely on vibrotactile and audio feedback from the walls during navigation.

The participants' reflections during the maze and post-maze show promising beginnings to meaningful discussions of animal empathy, as many drew connections to prior experiences of pets who were blind, deaf, or had their whiskers cut off and discussed how disorienting and difficult it would be for them to navigate with a sense removed. Participants described the whiskers as feeling like an extra sense, one that they were able to adapt to even in a short timeframe. Although losing their sight was disorienting, they were able to utilize the whiskers as a substitute for being able to "see." The combination of the Whisker Beard and maze activity suggests that through disorienting them and having them behave like a cat, they were able to consider what it would be like to be a cat relying on its whiskers every day, and how challenging it would be for a cat who has no whiskers at all.

## LIMITATIONS

### Hardware Implementation
The most severe technology design issue we encountered during the study was whisker placement; each participant noted that the lack of having a front-facing whisker on the forehead made it difficult to avoid obstacles directly in front of the wearer. Cats have a set of superciliary or suborbital whiskers on their brow for this very purpose, and the lack of these front-facing whiskers on our wearable was frustrating for participants. We did not include them in the original design because we wanted to focus on the whiskers on the sides of the faces, not realizing how important it would be to include them in other areas on the head.

### Study Design
Although participants were able to make use of the whiskers during this study, we believe that a longer study would give participants a more adequate amount of time to adapt to the

whiskers and therefore exhibit more natural behaviors, such as participants relying more on the whiskers than their hands during resource collection. It might also allow a different set of behaviors to emerge, as wearers become more comfortable in the environment and reduce thigmotaxic behavior.

## FUTURE WORK

We continue to investigate how the whisker wearable, and technologies like it, can remediate human experience in order to support deeper intersubjectivity with animals, and how such remediations can offer an experiential framework for science and science education.

### Iterative Design of the Wearable

To address the aforementioned technological limitations and move towards a more authentic design, we are working towards a more modular and customizable design. The next generation of the wearable will include additional sensing capabilities that support cat senses beyond whiskers, like hearing. A new custom board, Dr. Bones (Figure 10), will serve as a connection hub and accommodate the micro:bit as the primary controller for all input and output modules. Incorporating the micro:bit into our design will enable participants to program their individual modules using the Makecode programming environment. In exposing the individual sensing and output elements of this project we aim to encourage young people to create their own sensory augmentation systems. We conjecture that this customizability will offer a variety of ways for people to engage with scientific ideas relating to animals.

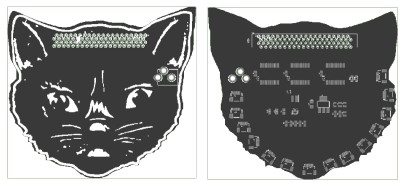

**Figure 10. Printed circuit board rendering of Dr. Bones.**

In addition to offering customizability for sensors, we will be offering people the ability to customize the placement of sensors and attachments on the wearable. For example, participants who wished they had a front-facing whisker during the activity will be able to create one. This will be more authentic to the way cats' whiskers work because cats and other mammals are able to direct control over the direction of their whiskers, as well as other parts of their bodies like their ears [46]. Further, we are addressing the lack of vibrotactile directional feedback by adding three motors per whisker to our design; therefore, through motor sequencing and drive intensity, we can render the direction and angle of the bend human sensation.

### Investigating Potential Experiences

One intriguing aspect of our work in adding whiskers to humans was the ability to create new types of empathetic experiences. While our initial work has focused on developing new technology and the analytical infrastructure needed to test it, we see several opportunities for more deeply exploring the experiential aspects of this technology. First, we may

consider how to explore the transhumanist aspects of gaining a new sensory capability. Can people find uses for wearable whiskers in their daily activities or as part of their everyday lived experience? By creating a more portable, lightweight version of the hardware, we may explore opportunities to send this device into the world to see what people make of it.

Second, we may consider how wearable whiskers can increase understanding of, and empathy for, the experiences of non-human animals. For example, by leading a person through an experience similar to the everyday behavior of a feral cat (sneaking through backyards, chasing birds, searching for edible items), can wearers of the whiskers better comprehend and empathize with the experiences of feral cats?

Finally, we may explore how experiences with wearable whiskers could increase an individual's understanding of, and relationship with, their own pets or other familiar animals. For example, children often must be taught what kinds of touch are liked and disliked by their pets; until they learn this, they may be frustrated by their pets' apparent distrust or fear when they are nearby. We may explore how to design experiences that can help a human understand a particular aspect of their pets' lived experience as a way of supporting a more respectful relationship between species.

## CONCLUSION

Wearable technologies and embodied learning experiences free humans from the confines of their biological limitations. This enables researchers to provide low-cost opportunities that offer firsthand perspective-taking experiences for people, allowing people to experiment with new sensory interactions, including ones that non-human animals have access to.

We presented the design of the Whisker Beard, which does just that—provides humans with the opportunity to experience what it would be like to have a new sense, in this case, what it would be like to have whiskers. We introduced concepts from animal behavioral science research and described how we applied it to evaluating the experiences of participants' while immersed in an animal perspective-taking activity. Our observations of participants' enactment of animal-like behaviors, as well as their impressions about the experience suggest that they were immersed in the sensory experience of being a cat with whiskers.

We are actively iterating on the designs of our hardware to offer more customizability. This will enable participants to design their own sensory augmenting technologies where they can explore their own curiosities about their pets' other senses. In near-future experiments we will iterate on the design of the wearable activity to offer a more immersive experience where participants can continue to enact animal-like behaviors. Our next steps will then be to investigate how participants developing increased awareness of animals' sensory experiences can support their enactment of empathetically-oriented design activities focused on improving animals' quality of life.

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
