# OpenReview forum: "Becoming Cat People: Animal-like Human Experiences with a Sensory Augmenting Whisker Wearable"
_graphicsinterface.org/Graphics_Interface/2020/Conference — Submitted to GI 2020_

### Official Review · AnonReviewer3 · 2020-01-06
**An intriguing study with good motivation and most of the procedures done right, but falls short on findings and does not deliver as promised.**

**Confidence:** 3
**Rating:** 4

**Review:**

In this paper, the authors present a whisker-like wearable device (Whisker Beard) and conducted a study to investigate how we can create technologies and environments that remediate non-human experiences (cat in this case), and human’s impressions towards them. Main findings include participants did exhibit navigation behaviour similar to whiskers-bearing animals, and drew connections to prior experiences with pets.

Overall I find this paper clear-written and original (I’m not aware of any studies like this). The study procedures are also well-explained. The motivation of enabling people to have an empathetic understanding of a situation through the sensory feelings of the affected animals is well-articulated and admirable. However, I think the significance/impact of the current stage of this work is lacking, and the findings don’t warrant the empathetic understanding that I believe would be the most valuable contribution of this work. Therefore I do not recommend accepting this paper.

Pros
-Good structuring of the paper.
-Made a good case of motivating this work with empathy and perspective-taking.
-Good visuals to illustrate results (though Figure 7 is just an instance of the analysis and the authors didn’t use it that much afterwards)
-Good summary of limitations on hardware implementation and study design, and what can be done to mitigate that.

Cons
-The linking between scientific investigation and empathy & perspective-taking is somewhat thin. I’m actually fine with just letting people “sense like their pet” and establish empathy on animals. In the same vein, relating this work with impairment also seems a bit of a stretch.
-The design/implementation of the whisker is unclear. Including details like the mapping between intensity of vibration and bend angles (not just they are “proportional”), where on the scalp (relative positions for each motor), on which development platform it is built (Arduino?) would make it better.
-Some findings are hard to verify (e.g., “some used their whiskers to gauge the width of the passage by moving back and forth brushing each side of their whiskers on the opposing walls.”) as there appears to have no other ways — participants were blindfolded and explicitly told to use their whiskers to navigate.
-The connection to prior experiences of pets drew by the participants feels like forced, because the researchers actually asked the participants this question.

As mentioned in the Cons, I find some the claims made in the Discussion not well-supported. As another example, the authors discussed sight being a dominant sense for obtaining information about an environment. So why did they blindfolded the participants? It does not make sense to strip that away and force people to rely in the whisker senses. A pair of tinted glasses or dimmed lighting would be more appropriate.

In the end, what this work really answered is just how human beings use a whisker-like sensory feature to navigate an unfamiliar environment, but not whether empathy is established nor the research questions being asked in the beginning of the paper.

---

### Official Review · AnonReviewer1 · 2020-01-08
**Borderline Accept: Seems principled but limited takeaways**

**Confidence:** 4
**Rating:** 6

**Review:**

# Summary

This paper describes the development of "Whisker Beard", a wearable interface that helps people shift perspectives to animals, such as cats. The device includes flex sensors attached to prosthetic whiskers, with flex values mapped to vibrotactile feedback on the scalp. Six participants were blindfolded and tasked with finding as many toy mice as possible (maximum of 10 were placed) in 10 minutes in a maze. Think aloud, video and focus group feedback was analyzed using qualitative coding techniques.

# Review

Overall, I find this paper is borderline but lean towards accept, as I don't have a strong reason to reject.

The positives: The design and methods seem principled and valid, it seems quite novel to me, and most of the related work is reasonably covered. There is some takeaway, as the authors report finding animal like behaviours from the participants, as well as some quotations that suggest a shifted perspective.

Limitations: I find that this paper could have several improvements that prevent me from more strongly arguing for accept.

1) The main concern I have is the limited takeaways. While the results seem valid, especially the coded results of how people behaved, I'm not really sure what to do with this information. Yes, it sounds like there is some evidence that this system can help people frame themselves as an animal, but I think the results would need more analysis before I would trust these as strong results.

2) I think the methods and results, while mostly valid, need additional analysis and reporting to be persuasive. The inter-rater reliability score is nice - it indicates that a principled coding process was followed - but I hoped to see more grounding in qualitative methodologies, which would help me trust the results of the coding process more. The quotations provided by participants are barely analyzed - simply sorted into lists - and here I think the results could be substantially stronger. The main goal of this system, if I understand the paper correctly, is to convey experience, not behaviour. As such, I think that a more thorough analysis of the participants' quotations, possibly using techniques like phenomenology and thick description, would really benefit the paper by interpreting and conveying participant experience. If this were done, then I could much more strongly argue that the paper should be accepted.

3) The system needs more description. I would have liked to see a full system diagram, and more description of the actuators used and how the voltage was mapped to vibrotactile feedback. I also would have liked to see a more rigorous calibration method - for this type of system, we don't need decibel values, but I would like to make sure the vibrotactile feedback is felt at similar levels of deflection for different participants, rather than just whether participants felt the feedback at all. (This is a minor point.)

4) The related work or design section should probably mention bio-inspired work on whisker sensors for robots, and whether those informed or inspired the work (or why not). Examples of such works: https://ieeexplore.ieee.org/abstract/document/220070, https://ieeexplore.ieee.org/abstract/document/1041717

Ultimately, after reading this paper - I'm intrigued, I think I've learned something, but I don't know how much I've learned. As such, I won't argue for rejection, but I cannot champion the paper.

---

### Official Review · AnonReviewer2 · 2020-01-09
**Reject: paper’s motivation is weak, research questions are too broad, the number of users is too low**

**Confidence:** 5
**Rating:** 4

**Review:**

The paper is written well and covers an interesting technology and experiment. The authors designed and tested an embodied experience in which they enable a human to navigate environments with artificial cat whiskers which can sense the environment. Paper raises interesting questions about replacing sensory experiences with wearables to better match those of animals, in an attempt to understand animals’ experiences. They had 6 participants navigate a small maze blindfolded while using the whiskers. They found similarities in behavior to the participants and animals in similar scenarios. The paper would be helpful for other researchers looking to explore embodied experiences.

Motivation
Embodied studies with pets is an intriguing and open problem, and there does seem to be a lack of papers that explore animals and especially pets in particular.
The related work section covers a lot of interesting and relevant information but is missing some important animal embodiment papers:
Arque: artificial biomimicry-lnspired tail for extending innate body functions (https://doi.org/10.1145/3305367.3327987)
A mobile pet wearable computer and mixed reality system for human–poultry interaction through the internet (https://doi.org/10.1007/s00779-005-0051-6)
The motivation to use cat whiskers specifically was not really there. It jumps directly in to whisker design and implementations. What other options did the authors consider that could enable this embodied experience? After all, RQ1 is "In what ways can we create technologies and environments that remediate human experiences to be like those of non-humans?"  I would either better motivate why cat whiskers was the chosen approach OR make RQ1 less general since the paper does not really dive in to any other approaches.
The study hinges on sensory deprivation but does not adequately address how that deviates from the “animal experience”, and most importantly how despite the eye-sight deprivation the reader should still be able to draw conclusions on the animal experience from the paper’s findings. As mentioned in the paper, eyesight is one of the predominant senses in cats as well. The lack of eyesight seems to heavily influence nearly every interaction in the study, which would call into question how relatable this task is to a general “cat-like” experience (except for perhaps a blind cat).
Motivation is too broad on pet embodiment, when it focuses on a particular pet and on a particular interaction.

Study:
In the paper, the authors write: "cats have difficulty seeing objects very close to their faces, so whiskers help them sense objects close to them, as well as protecting their face from harmful objects". The authors also mention that cats use both vision and whiskers to navigate. However, the study involves that people remove their sight by blindfolding them instead of alternatives such as limiting their vision.
For a short-term focused study, six (6) users seems like a small count. Results and discussion appear closer to a pilot study in their breadth and depth.

Clarity
Well-written.
Strong collection and range of cited works.

Originality
Very original and novel research. The experiment is interesting and the approach to analyzing the data is strong.
Focuses on increasing empathy.
The paper would be helpful for other researchers looking to explore embodied experiences.

Significance

Presented device seems well designed and appears to function well. Study and evaluation appear correct and sound. However, ultimately paper’s motivation is weak, research questions are too broad, the number of users is too low, and the study hinging on eyesight deprivation ultimately might influence the animal experience far too much for meaningful conclusions to be derived.


Recommendations
Make research questions less broad and more specific to what you actually did
Motivation needs to be MUCH stronger.
The Motivation mentions feminist critique and feminist science and education. It is unclear how these relate to the motivation at large and the project as a whole

- Recommend csquotes package for quotes instead of bullet list items (looks nicer)

- Add implications for how technology like this can benefit people.

---

### Meta-Review · Area_Chair1 · 2020-01-11

**Recommendation:** Reject
**Confidence:** 5

**Metareview:**

In this paper, the authors propose an interesting paper that explores what appears to be a novel wearable system for people to achieve a closer perspective to pets, specifically to cats in their particular system called Whisker Beard. The work involves the use of flex sensors to serve as prosthetic whiskers that are placed on the user’s face, which then provides vibrotactile feedback to the user’s scalp for better informing their surroundings as they are navigating.
The paper clearly brought consistency among the reviewers on their thoughts of the Whisker Beard system, particularly the strong and articulated writing style and the novelty of a wearable system that was designed for people to better empathize with their pets by attempting to emulate the physical sensory tools of a house cat. However, the reviewers also shared other concerns of the paper such as inadequate motivation, lack of implementation details, and weak study evaluation.
As a result, we believe that the paper is not in a ready-enough state to be accepted into this year's conference. Please see below for a list of the reviewers’ expanded explanation of the paper’s strengths and weaknesses. We believe that addressing these concerns will better prepare the work for submission to a future venue, and we wish the authors the best of luck in furthering improvements of this intriguing work.
Pros
1. The reviewers pointed out that the paper appeared to be original novel research.
2. The reviewers praised the paper for being well-written and strongly articulated.
3. With some exceptions, the reviewers felt that the literature review from the related works were thoroughly reviewed.
4. The reviewers expressed that the proposed wearable system seemed well-designed and operated as intended. They also shared that the study and evaluation appeared to be correct and sound.
5. The reviewers expressed that the authors provided a reasonable summary of the limitations on their wearable system's hardware implementation and study design, and followed up with the ways that can mitigate these limitations.

Cons
1. The reviewers had concerns regarding the planning and execution of the study, specifically the low count of six participants and the bias onto the participants from blindfolding them during the study.
2. The reviewers expressed some issues with the overly broad nature of the research questions due to the actual study of the paper not fully addressing those research questions.
3. Reviewers 2 and 3 communicated concerns regarding the wearable system lacking sufficient details on how it actually works.
4. Reviewers 1 and 2 expressed how the related work section was still lacking on including several prior studies that are similar to the study described in the paper.
5. Reviewers 1 and 3 stated their concerns on one execution aspect of the paper involving the blindfolding of the study participants. Specifically, the reviewers explained that the users' reliance on the whisker sensors did not seem to similarly reflect how cats use whiskers.
6. The reviewers had varying concerns regarding weak or lacking support of the paper's motivation claims.

---

### Decision · Program_Chairs · 2020-01-11

Reject